# Association between Recent Usage of Antibiotics and Immunogenicity within Six Months after COVID-19 Vaccination

**DOI:** 10.3390/vaccines10071122

**Published:** 2022-07-14

**Authors:** Ka-Shing Cheung, Lok-Ka Lam, Ruiqi Zhang, Poh-Hwa Ooi, Jing-Tong Tan, Wai-Pan To, Chun-Him Hui, Kwok-Hung Chan, Wai-Kay Seto, Ivan F. N. Hung, Wai K. Leung

**Affiliations:** 1Department of Medicine, Queen Mary Hospital, The University of Hong Kong, 102 Pokfulam Road, Hong Kong; cks634@hku.hk (K.-S.C.); llk719@ha.org.hk (L.-K.L.); zhangrq@hku.hk (R.Z.); rachelo2@hku.hk (P.-H.O.); tjt97@connect.hku.hk (J.-T.T.); twp856@ha.org.hk (W.-P.T.); hch309@ha.org.hk (C.-H.H.); wkseto@hku.hk (W.-K.S.); 2Department of Microbiology, Queen Mary Hospital, The University of Hong Kong, 102 Pokfulam Road, Hong Kong; chankh2@hku.hk

**Keywords:** COVID-19, vaccination, antibiotics, antibody, humoral

## Abstract

**Background:** Gut microbiota can be associated with COVID-19 vaccine immunogenicity. We investigated whether recent antibiotic use influences BNT162b2 vaccine immunogenicity. **Methods:** BNT162b2 recipients from three centers were prospectively recruited. Outcomes of interest were seroconversion of neutralising antibody (NAb) at day 21, 56 and 180 after first dose. We calculated the adjusted odds ratio (aOR) of seroconversion with antibiotic usage (defined as ever use of any antibiotics within six months before first dose of vaccine) by adjusting for covariates including age, sex, smoking, alcohol, and comorbidities. **Results:** Of 316 BNT162b2 recipients (100 [31.6%] male; median age: 50.1 [IQR: 40.0–57.0] years) recruited, 29 (9.2%) were antibiotic users. There was a trend of lower seroconversion rates in antibiotic users than non-users at day 21 (82.8% vs. 91.3%; *p* = 0.14) and day 56 (96.6% vs. 99.3%; *p* = 0.15), but not at day 180 (93.3% vs. 94.1%). A multivariate analysis showed that recent antibiotic usage was associated with a lower seroconversion rate at day 21 (aOR 0.26;95% CI: 0.08–0.96). Other factors associated with a lower seroconversion rate after first dose of the BNT162b2 vaccine included age ≥ 60 years (aOR: 0.34;95% CI: 0.13–0.95) and male sex (aOR: 0.14, 95% CI: 0.05–0.34). There were no significant factors associated with seroconversion after two doses of BNT16b2, including antibiotic use (aOR: 0.03;95% CI: 0.001–1.15). **Conclusions:** Recent antibiotic use may be associated with a lower seroconversion rate at day 21 (but not day 56 or 180) among BNT162b2 recipients. Further long-term follow-up data with a larger sample size is needed to reach a definite conclusion on how antibiotics influence immunogenicity and the durability of the vaccine response.

## 1. Introduction

As of March 2022, COVID-19 has affected more than 480 million people and caused more than six million deaths worldwide. Among different measures to contain the spread of the virus, vaccination is of paramount importance in preventing infection, severe symptoms, and death [1]. Currently available COVID-19 vaccines have been developed via different technology platforms including the inactivated virus, RNA-based, adenovirus vector and protein subunit types [2]. The two currently available vaccines in Hong Kong are BNT162b2 (BioNTech; mRNA vaccine) and CoronaVac (Sinovac Biotech; inactivated virus vaccine). The overall efficacy (protection from symptomatic COVID-19) after two doses of BNT162B2 is 95% [3], while that of CoronaVac is 70% [4]. In a Hong Kong study, BNT162b2 induced an approximately 10-fold higher neutralizing antibody level than CoronaVac [5].

Younger age and female sex are associated with higher vaccine efficacy and adverse reactions for influenza [6]. Overweight and obesity is a risk factor for poorer immune response for influenza vaccination [7], and therefore possibly COVID-19 vaccine immunogenicity. The composition and function of gut microbiota are increasingly recognised to play an important role in modulating immune response to various kinds of vaccination [8], including influenza vaccine response [6]. A study showed that antibiotic-induced gut microbiota perturbation could lead to suboptimal antibody production and reduced antibody affinity among those with low pre-existing antibody titers against influenza virus [9]. A few randomized controlled trials (RCT) showed that probiotics (including Lactobacillus plantarum [10], Lactobacillus GG [11], and Bifidobacterium animalis ssp [12]) induce higher immunoglobulin levels against certain influenza strains after intranasal or parental influenza vaccination.

As age, sex, overweight/obesity are associated with vaccine efficacy against SARS-CoV-2 as in influenza [13], other factors that influence influenza vaccine response may be applicable to COVID-19 vaccination. Intuitively, antibiotic use could reduce COVID-19 vaccine immunogenicity via gut dysbiosis. However, research on this aspect is currently lacking.

Studies have shown that neutralizing antibody levels are surrogate markers of vaccine effectiveness [14], which are predictive of protection from symptomatic COVID-19 infection [15,16]. Therefore, this prospective cohort study aimed to determine the potential association between the recent use of antibiotics and vaccine immunogenicity and reactogenicity among COVID-19 vaccine recipients.

## 2. Methods

### Study Design and Participants

This is a prospective multi-center cohort study recruiting adult subjects receiving BNT162b2 vaccine from three vaccination centers in Hong Kong (Sun Yat Sen Memorial Park Sports Centre, Ap Lei Chau HKU Vaccination Centre and Queen Mary Hospital). Exclusion criteria included age < 18 years, history of gastrointestinal surgery, inflammatory bowel disease, immunocompromised status including post-transplantation and immunosuppressives/chemotherapy, other medical diseases (cancer, hematological, rheumatological and autoimmune diseases), those with prior COVID-19 infection (identified from history taking or presence of antibodies to SARS-CoV-2 nucleocapsid (N) protein. This antibody is not inducible by SARS-CoV- vaccines and therefore is an indicator of past infection. 

Two doses of intramuscular BNT162b2 (0.3 mL) were administered at three weeks apart as recommended. Their blood samples were collected at four time-points: (i) before vaccination (baseline); (ii) 21 days after first dose; (iii) 56 days after first dose; and (iv) 180 days after first dose. A prior study showed that BNT162b2 protection against infection peaked in the first month after the second dose [17]. The blood samples were tested for neutralizing antibody (NAbs) SARS-CoV-2 receptor-binding domain (RBD) [18]. NAb seroconversion was defined as 15 AU/mL.

Subjects recorded any adverse reactions daily for seven days after vaccination, including systemic reactions (fever, chills, headache, tiredness, nausea, vomiting, diarrhoea, myalgia, arthralgia, and skin rash) and local reactions (pain, erythema, swelling, ecchymosis and itchiness). The severity of reactions were graded as 1, 2, 3, and 4, according to toxicity grading scale by the U.S. Department of Health and Human Services [19].

The study was approved by the Institutional Review Board of the University of Hong Kong (HKU) and Hong Kong West Cluster (HKWC) of Hospital Authority. All patients provided written informed consent for participation in this study.

## 3. Outcomes of Interest

Primary outcomes of interest were seroconversion rates at three time points after the first dose of BNT162b2 (day 21, 56, and 180). Testing for NAb against SARS-CoV-2 RBD was performed using the new version of the iFlash-2019-nCoV Nab kit (chemiluminescent microparticle immunoassay; Shenzhen YHLO Biotech Co, Ltd., Shenzhen, China) [18]. Briefly, serum samples were placed on a sample rack in the sample loading area, and a reagent pack with 2019-nCoV RBD antigen (30KD)-coated paramagnetic microparticles and acrodinium ester-labelled ACE2 conjugate were placed in the reagent loading area. The iFlash system performed all functions automatically and measured the signal from the chemiluminescent reaction. Values between >9 and <15 AU/mL were regarded as “indeterminate” results. The lower limit was set to <4 AU/mL and the upper limit was set to >800 AU/mL.

A secondary outcome of interest was reactogenicity, defined as adverse reactions within seven days of each injection [20].

## 4. Exposure of Interest and Covariates

Exposure of interest was pre-vaccination antibiotic use, defined as ever use of any antibiotics within six months before vaccination [21]. The six-month cut-off was used because antibiotic-induced gut microbiota perturbation took months to recover [9]. The antibiotics included 11 classes of antibiotics, which were penicillins, cephalosporins, quinolones, tetracyclines, carbapenems, macrolides, aminoglycosides, glycopeptides, nitroimidazoles, sulpha/trimethoprim, and other antibiotics (daptomycin, clindamycin, linezolid, nitrofurantoin, rifaximin, and rifampicin). Appendix A shows the classification of antibiotics based on the anti-bacterial spectrum.

Other covariates taken into analysis were age (cutoff of 60 years) [13], sex, diabetes mellitus (DM) [22], overweight (BMI ≥ 23 kg/m^2^) [13,23], hypertension, raised LDL (≥3.4 mmol/L), moderate-to-severe hepatic steatosis (defined as controlled attenuated parameter [CAP] ≥ 268 dB/M on transient elastography), smoking, and alcohol [22].

## 5. Statistical Analyses

All statistical analyses were performed using R version 3.2.3 (R Foundation for Statistical Computing) statistical software. Continuous variables were expressed as the median and interquartile range (IQR). A Mann-Whitney U-test was used to compare the continuous variables of the two groups. A Chi-square test or Fisher’s exact test was applied for categorical variables. A multivariable logistic regression model was applied to derive the adjusted odds ratio (OR) of seroconversion at the aforementioned time points for BNT162b2 recipients.

Several sensitivity analyses was performed. First, proton pump inhibitors (PPI) use was included into the multivariable analysis, as PPIs are known to affect gut microbiota [24], yet there is currently a lack of evidence that PPIs reduce vaccine immunogenicity. Second, antibiotic use was defined by varying duration of antibiotic usage within the six-month period before first dose of vaccination (≥one week and ≥one month). Third, antibiotic use was defined by varying the time of last antibiotic usage before first dose of vaccination (within three months, one month, two weeks and one week).

A two-sided *p*-value of <0.05 was regarded to be statistically significant.

## 6. Results

### 6.1. Patient Characteristics

A total of 316 BNT162b2 recipients were recruited (Table 1). On hundred (31.6%) were male and the median age was 50.1 years (IQR: 40.0–57.0). At day 56, 284 (89.9%) subjects had their neutralising antibody levels measured. At day 180, 185 (58.5%) BNT162b2 subjects had available neutralising antibody data.

There were 29 (9.2%) antibiotic users and 287 (90.8%) antibiotic non-users. The median duration of antibiotic use was 7 days (IQR: 7–13). The median time of last antibiotic use from baseline was 70 days (IQR: 33–135). There was no significant difference in the baseline characteristics between antibiotic users and non-users (Table 1). The most common indication for antibiotics was skin and soft tissue infection (*n* = 15), followed by dental infection (*n* = 6) and genitourinary infection (*n* = 4). Other indications of antibiotics included gastroenteritis (*n* = 1), Helicobacter pylori infection (*n* = 1), upper respiratory tract infection (*n* = 1), and unknown (*n* = 1) (Table 2). All antibiotic users had received broad-spectrum antibiotics, with 26 using either amoxycillin/clavulanic acid or amoxycillin (four of whom had concomitant metronidazole use), one used metronidazole alone, one used doxycycline alone, and one used levofloxacin alone.

### 6.2. Humoral Immune Response among BNT162b2 Recipients

After one dose of BNT162b2, there was a non-significant trend in the seroconversion rate (82.8% vs. 91.3%; *p* = 0.14) and the median antibody level (31.5 vs. 36.2 AU/mL; *p* = 0.68) between antibiotic users and non-users. After two doses of BNT162b2, the non-significant trend remained but diminished for the seroconversion rate (96.6% vs. 99.3%; *p* = 0.15) and median antibody level (596.1 vs. 668.6 AU/mL; *p* = 0.11) (Figure 1).

Factors associated with the seroconversion of neutralising antibodies at day 21 after one dose of BNT162b2.

Independent factors associated with seroconversion of neutralising antibody after one dose of BNT162b2 included antibiotic use (aOR: 0.26, 95% CI: 0.08–0.96), age ≥ 60 years (aOR: 0.34, 95% CI: 0.13–0.95) and male sex (aOR: 0.14, 95% CI: 0.05–0.34) (Table 3), but not other cardiometabolic risk factors including DM.

A sensitivity analysis was undertaken by including PPIs into the multivariable analysis, and showed that the aOR of seroconversion was 0.26 (95% CI: 0.08–0.96) with antibiotic use and 1.13 (95% CI: 0.30–5.84) with PPI use. Sensitivity analyses with different definitions of antibiotic use by either varying the duration of usage within six months before first dose of vaccination or varying the time of last antibiotic usage before first dose of vaccination did not reveal any statistical significance, likely due to underpower (Appendix A). 

Factors associated with seroconversion of the neutralising antibody at day 56 after two doses of BNT162b2. There were no significant factors associated with seroconversion after two doses of BNT16b2 including antibiotic use (aOR: 0.03, 95% CI: 0.001–1.15) (Table 3). 

### 6.3. Reactogenicity among BNT162b2 Recipients

Two hundred and eighty (88.6%) BNT162b2 recipients reported adverse effects within seven days of either dose of vaccine (Appendix A). All adverse effects were mild (grade 1 and 2) and self-limiting, with no serious adverse events (grade 3 and 4) such as anaphylaxis or cardiovascular events reported. The most common local and systemic adverse reactions were injection site pain (84.8%) and fatigue (47.5%).

There was no significant difference in frequency of adverse reactions between antibiotic users and non-users (any reaction: 25 [86.2%] vs. 255 [88.9%], *p* = 0.67; local reaction: 24 [82.8%] vs. 248 [86.4%], *p* = 0.59; systemic reaction: 16 [55.2%] vs. 179 [62.4%], *p* = 0.45). A lower rate of systemic adverse reaction of vomiting was observed in antibiotic non-users (2 [6.9%] vs. 4 [1.4%], *p* = 0.04).

## 7. Discussion

In this prospective cohort study involving 316 BNT162b2 vaccine recipients, we showed that the recent usage of antibiotics, within six months before vaccination, was associated with a 74% lower rate of seroconversion of neutralizing antibodies after one dose of BNT162b2. However, recent antibiotic usage was not associated with impaired vaccine immunogenicity after two doses of BNT162b2. 

There are several potential mechanisms by which microbiota can modulate vaccine immunogenicity, including the production of immunomodulatory molecules, the regulation of production of immunomodulatory cytokines, the production of immunomodulatory metabolites and the encoding of epitopes that are cross-reactive with vaccine-encoded epitopes [8]. Previous animal studies have linked the importance of commensal microbiota to immunity against influenza virus, which was severely impaired in germ-free or antibiotic-treated mice when compared with specific pathogen-free mice [25,26]. Gut microbiota has been shown to regulate the immunity in the respiratory tract mucosa via regulation of toll-like receptor 7 (TLR7), signalling a pathway for activation of inflammasomes [27]. TLR5-mediated sensing of flagellin produced by gut microbiota is critical to restoration of antibody response after influenza vaccination in germ-free or antibiotic-treated mice [28]. Antigen presenting cells like dendritic cells present vaccine antigens to T cells and secrete immunomodulatory cytokines. The microbiota regulates production of type I interferons by plasmacytoid dendritic cells, which in turn affects the conventional dendritic cells functioning on T cell priming [29]. Microbiota-derived short chain fatty acids (SCFAs), e.g., acetate, butyrate and propionate, also play a role in homeostatic and pathogen-specific antibody response [30]. SCFAs can enhance B cell metabolism for increased antibody production and enhance the expression of genes crucial for plasma cell differentiation and class switching [8]. B cell or T cell cross-reactivity with microbiota-derived and vaccine-encoded epitopes may also alter vaccine immune response [31,32]. Therefore, it is plausible that gut microbiota also have bearings on vaccine immunogenicity against SARS-CoV2 also.

A clinical study showed that a five-day broad spectrum antibiotic regimen consisting of neomycin, vancomyin and metronidazole led to gut dysbiosis [9]. There was a predominance of Enterobacteriaceae and Streptococcaceae at early time points and the diminished abundance of Lachnospiraceae, Runinococcaceae, Bacteroidaceae and Veillonellaceae. This disturbance in bacterial relative abundance returned to baseline only three months after the antibiotic regimen. Both the alpha (within-sample) and beta (between-sample) diversity of gut microbial composition of the antibiotic group did not fully recovered even six months after antibiotic treatment. This antibiotic-induced gut microbiota perturbation led to suboptimal impairment in influenza H1N1-specific neutralization and binding IgG1 and IgA responses among those with low pre-existing antibody titers [9]. Furthermore, antibiotics reduced serum secondary bile acids and enhanced the expression of inflammatory signatures (including AP-1/NR4A expression). 

Our current study is the first to report an association between the usage of antibiotics and impaired early vaccine immunogenicity after one dose of BNT162b2. The seroconversion rate of BNT162b2 was 82.8% and 91.3% for antibiotic users and non-users, respectively. Although the comparison was statistically non-significant (*p* = 0.14), underpower may be an issue due to the low number of patients with recent antibiotics use. Nonetheless, after multivariable analysis including potential variables that may affect vaccine response (e.g., age, sex, DM), we found that antibiotic usage was associated with a 74% lower chance of seroconversion after one dose of BNT162b2. All antibiotics users had broad-spectrum antibiotics, and therefore the effect of narrow-spectrum antibiotics could not be studied in the current study. Similarly, on day 56, a trend towards a lower seroconversion rate was observed for antibiotic users versus non-users (96.6% vs. 99.3%; *p* = 0.15) with an OR of 0.03 (95% CI: 0.001–1.15).

As this is a prospective cohort study, it is less likely to suffer from recall bias and interviewer bias in terms of antibiotic use. Our results shed light on the potential interaction between gut microbiota and vaccine immunogenicity. It also highlights that subjects who have received antibiotics within six months before vaccination should not delay the receiving of a second dose, as they are less likely to mount a sufficient antibody response with one dose of vaccination.

Previous studies found that extending the interval between the first and second dose of the vaccine to 8–12 weeks for mRNA vaccines may enhance the serologic response [33,34] and hence lead to better long-term protection [15]. However, both antibiotic users and non-users achieved high seroconversion rates (>96%) after two doses of BNT162b2.

Other risk factors for negative serological response after one dose of BNT162n2 included male sex and older age. Male sex was also found to be a risk factor for impaired vaccine response in activated whole virion vaccines among patients with chronic liver diseases [35]. Possible reasons are testosterone-modulating genes mediating metabolism of lipid and neutralizing antibody response, as in influenza vaccination [36]. Although PPIs affect gut microbiome by decreasing the diversity, increasing the relative abundance of multiple oral bacteria, *Enterococcus, Streptococcus, Staphylococcus* and pathogenic species *Escherichia coli* [24]. Nevertheless, there is currently a lack of evidence that PPIs reduce vaccine immunogenicity. Our study also did not reveal an effect of PPIs on vaccine immunogenicity, although further studies with larger sample sizes should focus on the potential unfavourable of PPIs on vaccine immune response.

One of the limitations of the current study is the definition of antibiotic use, including the duration of exposure and the time interval between last antibiotic use and vaccination. We used an arbitrary cutoff of ever use of any antibiotics within six months before vaccination [21], as antibiotic-induced gut microbiota perturbation took at least six months to recover [9]. Due to the relatively small number of antibiotic users (*n* = 29), any stratified analysis according to duration of antibiotic use and the time interval from last antibiotic use will likely suffer from underpower. The relatively small sample size of this cohort also precludes stratified analysis according to different variables.

Other limitations should be noted. First, confounding by indication of antibiotics may exist, in which those who received antibiotics may have an underlying impaired immune system. However, this bias was minimized by the exclusion of study subjects with cancer, rheumatological/autoimmune diseases, and those receiving immunosuppressives/chemotherapy. Table 1 also shows very similar baseline characteristics between antibiotic users and non-users. The absence of a difference in seroconversion rates between two doses of BNT162b2 provided further evidence against confounding by indication. Second, gut microbiota data among the vaccine recipients were not available and, hence, we could not confirm the postulation that antibiotics impair early vaccine immunogenicity via its effects on gut microbiota. Identifying which bacterial species are associated with COVID-19 vaccine immunogenicity may have therapeutic implications in terms of use of probiotics to enhance and maintain vaccine immunogenicity. Third, we only performed the antibody assay against wild type viruses. The vaccine immunogenicity was shown to be weaker against the Delta variant [37] and Omicron variant [38], and hence our study results were not generalizable to other mutant strains. Fifth, although antibodies are important markers of vaccine protection, our study did not have data on the cellular immune response, which also plays an important role in protection against COVID-19 [39,40], in particular severe infection. Lastly, data on the persistence of the antibody levels at one year after vaccination were not available yet. Hence, we could not study whether antibiotics will affect the persistence of serum neutralizing antibody levels post-vaccination. There are reports of COVID-19 re-infection three to five months after the initial infection due to the diminishing serum neutralizing antibody levels [41,42]. A recent study showed that vaccine effectiveness against COVID-19 infection progressively waned–from 92% at 15–30 days to 47% at 121–180 days and 23% from day 211 onwards among BNT162b2 recipients [43].

## 8. Conclusions

Recent antibiotic use may be associated with a lower seroconversion rate at day 21 (but not day 56 or 180) among BNT162b2 recipients. Further long-term follow-up data with a larger sample size and a stratified analysis according to duration of antibiotic exposure and time interval from last antibiotic use will be helpful to reach a definite conclusion and to further delineate how antibiotics influence immunogenicity and the durability of the vaccine response.

## Figures and Tables

**Figure 1 vaccines-10-01122-f001:**
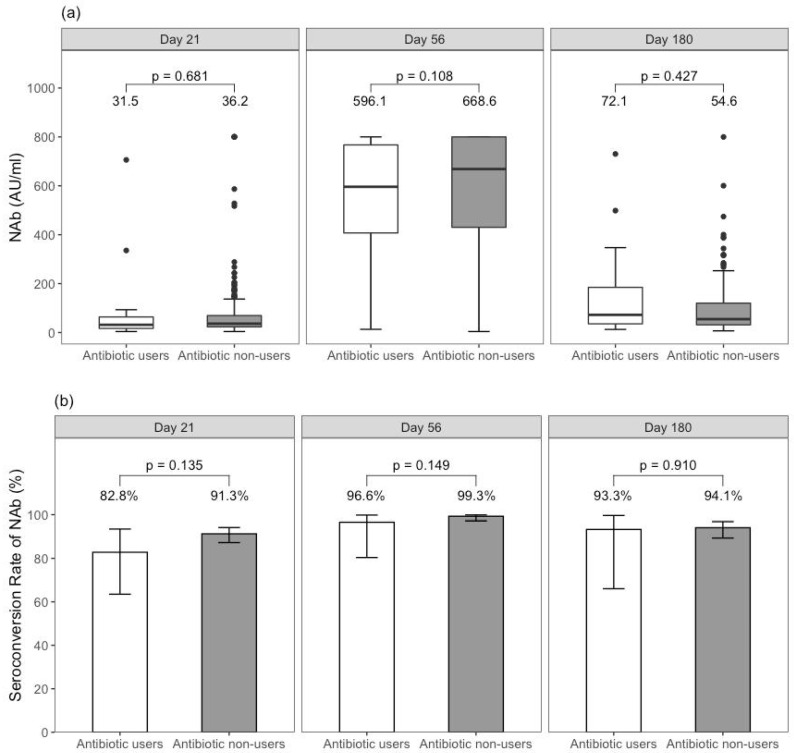
Comparison of (**a**) neutralising antibody level and (**b**) seroconversion rate between antibiotic users and non-users among BNT162b2 recipients. Note: 316 subjects had neutralising antibody level measured at day 21, 284 subjects at day 56, 185 subjects at day 180.

**Table 1 vaccines-10-01122-t001:** Baseline characteristics between antibiotic users and non-users among BNT162b2 recipients.

	All (*n* = 316)	Antibiotic Users (*n* = 29)	Antibiotic Non-Users (*n* = 287)	*p*-Value
Age ≥ 60 years (*n*, %)	47 (14.9%)	3 (10.3)	44 (15.3)	0.472
Male sex (*n*, %)	100 (31.6%)	6 (20.7)	94 (32.8)	0.183
DM (*n*, %)	23 (7.3%)	4 (13.8)	19 (6.6)	0.157
Overweight/obesity (*n*, %)	156 (49.4%)	14 (48.3)	142 (49.5)	0.902
Hypertension (*n*, %)	98 (31.0%)	6 (20.7)	92 (32.1)	0.207
Raised LDL (≥3.4 mmol/L) (*n*,%)	57 (18.0%)	9 (31.0)	48 (16.7)	0.056
Smoking (*n*, %)	14 (4.4%)	0 (0)	14 (4.9)	0.224
Alcohol use (*n*, %)	18 (5.7%)	1 (3.4)	17 (5.9)	0.584
Moderate/severe hepatic steatosis (CAP ≥ 268 dB/M) (*n*, %)	70 (22.2%)	8 (27.6)	62 (21.6)	0.460

Abbreviations: DM, diabetes mellitus; LDL, low density lipoprotein; CAP, controlled attenuated parameter.

**Table 2 vaccines-10-01122-t002:** Antibiotic use details and concomitant proton pump inhibitor use.

Patient	Indication of Antibiotic Use	Type of Antibiotics	Dosage	Total Duration	PPI Use
1	Skin and soft tissue infection	Amoxycillin/clavulanic acid	1 g BD	14 days	0
2	Skin and soft tissue infection	Amoxycillin/clavulanic acid	1 g BD	7 days	0
3	Skin and soft tissue infection	Amoxycillin/clavulanic acid	1 g BD	12 days	0
4	Skin and soft tissue infection	Amoxycillin/clavulanic acid	375 mg tds	7 days	14 days
5	Skin and soft tissue infection	Amoxycillin/clavulanic acid	375 mg tds	7 days	0
6	Skin and soft tissue infection	Amoxycillin/clavulanic acid	1 g BD	19 days	0
7	Skin and soft tissue infection	Amoxycillin/clavulanic acid	1 g BD	7 days	42 days
8	Skin and soft tissue infection	Amoxycillin/clavulanic acid	1 g BD	13 days	0
9	Skin and soft tissue infection	Amoxycillin/clavulanic acid	1 g BD	7 days	0
10	Skin and soft tissue infection	Amoxycillin/clavulanic acid	375 mg tds	7 days	0
11	Skin and soft tissue infection	Amoxycillin/clavulanic acid	1 g BD	15 days	0
12	Skin and soft tissue infection	Doxycycline	100 mg daily	91 days	0
13	Skin and soft tissue infection	Amoxycillin/clavulanic acid	375 mg tds	7 days	0
14	Skin and soft tissue infection	Amoxycillin/clavulanic acid	375 mg tds	5 days	35 days
15	Skin and soft tissue infection	Amoxycillin/clavulanic acid	375 mg tds	7 days	0
16	Dental infection	Amoxycillin	NA	3 days	0
17	Dental infection	Amoxycillin + Metronidazole	500 mg tds 400 mg tds	15 days	0
18	Dental infection	Amoxycillin/clavulanic acid	375 mg tds	7 days	14 days
19	Dental infection	Amoxicillin + metronidazole	250 mg tds 400 mg tds	7 days	0
20	Dental infection	Amoxycillin/clavulanic acid + metronidazole	375 mg tds 200 mg tds	4 days	0
21	Dental infection	Metronidazole	200 mg tds	7 days	0
22	Genitourinary tract infection	Amoxycillin/clavulanic acid	1 g BD	7 days	0
23	Genitourinary tract infection	Amoxycillin/clavulanic acid	1 g BD	7 days	0
24	Genitourinary tract infection	Amoxycillin/clavulanic acid	1 g BD	7 days	0
25	Genitourinary tract infection	Amoxycillin/clavulanic acid	1 g BD	7 days	0
26	Gastroenteritis	Amoxycillin/clavulanic acid + metronidazole	NA	10 days	0
27	*Helicobacter pylori* infection	Amoxycillin + Clarithromycin	1 g BD 500 mg BD	14 days	14 days
28	Upper respiratory tract infection	Amoxycillin	NA	14 days	49 days
29	Not documented	Levofloxacin	500 mg daily	7 days	0

Abbreviations: BD, twice a daily; tds, three times a day.

**Table 3 vaccines-10-01122-t003:** Risk factors for the seroconversion of neutralizing antibody among BNT162b2 recipients.

	Adjusted OR *	95% CI
**BNT162b2 (one dose)** **Day 21**		
Antibiotic usage	0.26	0.08–0.96
Age ≥ 60 years	0.34	0.13–0.95
Male sex	0.14	0.05–0.34
DM	0.42	0.12–1.66
Overweight/obesity	0.92	0.37–2.29
Hypertension	1.07	0.44–2.82
Raised LDL (≥3.4 mmol/L)	3.82	1.01–25.4
Smoking	1.14	0.22–9.40
Alcohol use	1.02	0.26–5.20
Moderate/severe hepatic steatosis (CAP ≥ 268 dB/M)	2.48	0.80–8.83
**BNT162b2 (two doses)** **Day 56**		
Antibiotic usage	0.03	0.001–1.15
Age ≥ 60 years	0.38	0.006–69.18
Male sex	NA *	NA *
DM	0.02	0.0002–2.78
Overweight/obesity	0.58	0.008–27.85
Hypertension	0.40	0.008–25.06
Raised LDL (≥3.4 mmol/L)	0.56	0.01–101.48
Smoking	NA *	NA *
Alcohol use	NA *	NA *
Moderate/severe hepatic steatosis (CAP ≥ 268 dB/M)	NA *	NA *

* adjusted OR and 95% CI could not be derived due to the absence of negative serological response among all female patients, smokers, those who used alcohol and those with moderate/severe hepatic steatosis. Abbreviations: OR, odds ratio; 95% CI, 95% confidence interval; DM, diabetes mellitus; LDL, low density lipoprotein; CAP, controlled attenuated parameter.

## Data Availability

Data sharing is not available due to confidentiality issue.

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
