# Peer review of "Association between Recent Usage of Antibiotics and Immunogenicity within Six Months after COVID-19 Vaccination"

_vaccines, 2022, doi:10.3390/vaccines10071122_

Round 1

Reviewer 1 Report

Dear Authors,

I have read this manuscript with great interest. The underlying idea can be relevant, depending on the results.

I have some comments, questions and suggestions to try to improve the text.

1) Some interesting information is given in the supplementary tables. For example, the antibiotics used or the indication of use of these antibiotics.

In my opinion, this information should be presented in the results and appropriately discussed. It should be clearly ruled out the possibility that the infection causing the prescription of an antibiotic was not the cause of the lower immunological response.

For example, 2 patients probably received omeprazole ("gastrointestinal infection" and "H. pylori"). And omeprazole is not an antibiotic but has deep effects on gut microbiota. In one additional case, there was no information. This makes 7% of the patients treated with antibiotics, probably had other causes not considered in the study which could be associated with gut microbiota alterations. What happened with the remaining patients?

2) "Anti anaerobic" antibiotics is a disputed term (Woerther PL, et al. Is the term “anti-anaerobic” still relevant? Int J Infect Dis 2021; 102: 178-180)

3) Maybe previous articles have focused on this topic. It would be interesting to cite them. For example Lynn, DJ et al. Modulation of immune responses to vaccination by the microbiota: implications and potential mechanisms. Nat Rev Immunol 2022; 22, 33–46

4) What about the duration of antibiotic treatments? This factor should be taken into account, described in the results and commented on in the discussion. Taking antibiotics should not be a "yes/no" situation. The impact is different taking one day, one week or one month.

5) As it is said in the manuscript, the gut microbiome takes time to come back to normal. I agree on the 6-month exposition window. But, again, it should not be a "yes/no" question. It is different if the exposure was 6 months before receiving the vaccine or if it took place two weeks before the jab. This should be described, analysed and discussed.

I'm aware that the sample size is small, especially the cohort that took antibiotics. So, it is difficult to stratify according to different variables. But perhaps just considering "exposure during the previous 6 months" is far too generic to conclude anything.

In my opinion, all these points should be addressed to give the most accurate information to the potential reader. Perhaps the conclusions, both in the Abstract and the final section of the manuscript should also be modified accordingly.

Reviewer 2 Report

Title

ASSOCIATION BETWEEN RECENT USAGE OF ANTIBIOTICS AND EARLY COVID-19 VACCINE IMMUNOGENICITY

-the title needs modification: early C-19 vaccine immunogenicity sounds ambiguous

Abstract

Background: Gut microbiota can be associated with COVID-19 vaccine immunogenicity. We investigated whether recent antibiotics use would influence immunogenicity of BNT162b2 vaccine. Methods: BNT162b2 recipients from three vaccination centers were prospectively recruited. Outcomes of interest were seroconversion of neutralising antibody (NAb) at day 21, 56 and 180 after first dose. We calculated the adjusted odds ratio (aOR) of seroconversion with antibiotic usage (defined as ever use of any antibiotics within 6 months before first dose of vaccine) by adjusting for covariates including age, sex, smoking, alcohol, and underlying comorbidities. Results: Of 316 BNT162b2 recipients (100 [31.6%] male; median age:50.1 [IQR:40.0-57.0] years) recruited, 29 (9.2%) were antibiotic users. There was a trend of lower seroconversion rates in antibiotic users than non-users at day 21 (82.8% vs 91.3%; p=0.14) and day 56 (96.6% vs 99.3%, p=0.15), but not at day 180 (93.3% vs 94.1%). Multivariate analysis showed that recent antibiotic usage was associated with lower seroconversion rate at day 21 (aOR 0.26; 95% CI:0.08-0.96). Other factors associated with lower seroconversion rate after first dose of BNT162b2 vaccine included age >60 years (aOR:0.34, 95% CI:0.13–0.95) and male sex (aOR:0.14, 95% CI:0.05–0.34). There were no significant factors associated with seroconversion after two doses of BNT16b2, including antibiotic use (aOR:0.03, 95% CI:0.001–1.15). Conclusion: Recent antibiotic use was associated with a higher risk of negative serological response after the first dose of BNT162b2, but not second dose.

Comment: the conclusion does not seem to be in congruent with the findings

2. METHODS

2.1. Study design and participants

2.2. Outcome of interest

2.3. Exposure of interest and covariates

2.4. Statistical analyses

3. Results

3.1. Patient characteristics

3.2. Humoral immune response among BNT162b2 recipients

3.3. Reactogenicity among BNT162b2 recipients

Methods and Results are coherent and easy to follow

Suggest to definite the type, dosage and indication of antibiotic use

Figure 1. Comparison of (a) neutralising antibody level and (b) seroconversion rate between antibi-152 otic users and non-users among BNT162b2 recipients. Note: 316 subjects had neutralising antibody 153 level measured at day 21, 284 subjects at day 56, 185 subjects at day 180

-error bar should be included

Table 1. Baseline characteristics between antibiotic users and non-users among BNT162b2 recipients

Table 2. Risk factors for seroconversion of neutralizing antibody among BNT162b2 recipients.

Table 3. Association between antibiotics and seroconversion of neutralizing antibody according to 166 nature of antibiotics after one dose of BNT162b2.

Table 1 -3 are well organised

5. CONCLUSION  

Use of antibiotics within 6 months before BNT162b2 vaccination was associated with  a lower seroconversion rate after a single dose of vaccine, but not two doses. Further re-268 search on the association among antibiotics, gut microbiota and COVID-19 vaccine immunogenicity is warranted

Comment: the conclusion does not seem to be in congruent with the findings 

Round 2

Reviewer 1 Report

Dear Authors,

Thank you for your work on this new version of the manuscript. In my opinion, the results are clearer now. And the analyses cover possible questions by readers and describe the study's limitations.

Kind regards